# Integrating Phylogeographic Analysis and Geospatial Methods to Infer Historical Dispersal Routes and Glacial Refugia of *Liriodendron chinense*

**Yufang Shen** [1], **Yanli Cheng** [1], **Kangqin Li** [1,2] and **Huogen Li** [1,*]

1   Key Laboratory of Forest Genetics & Biotechnology of Ministry of Education, Co-Innovation Center for
    Sustainable Forestry in Southern China, Nanjing Forestry University, Nanjing 210037, China
2   Jiangxi Academy of Forestry, Nanchang 330032, China
*   Correspondence: hgli@njfu.edu.cn; Tel.: +86-025-8542-8731

**Abstract:** *Liriodendron chinense* (Hemsl.), a Tertiary relic tree, is mainly distributed in subtropical China. The causes of the geographical distribution pattern of this species are poorly understood. In this study, we inferred historical dispersal routes and glacial refugia of this species by combining genetic data (chloroplast DNA (cpDNA), nuclear ribosomal DNA (nrDNA), and nuclear DNA (nDNA)) and geospatial data (climate and geology) with the methods of landscape genetics. Additionally, based on sequence variation at multiple loci, we employed GenGIS and Barrier software to analyze *L. chinense* population genetic structure. Dispersal corridors and historical gene flow between the eastern and western populations were detected, and they were located in mountainous regions. Based on species distribution model (SDMs), the distribution patterns in paleoclimatic periods were consistent with the current pattern, suggesting the presence of multiple refuges in multiple mountainous regions in China. The genetic structure analysis clustered most eastern populations into a clade separated from the western populations. Additionally, a genetic barrier was detected between the eastern and western populations. The dispersal corridors and historical gene flow detected here suggested that the mountains acted as a bridge, facilitating gene flow between the eastern and western populations. Due to Quaternary climatic fluctuations, the habitats and dispersal corridors were frequently inhabited by warm-temperate evergreen forests, which may have fragmented *L. chinense* habitats and exacerbated the differentiation of eastern and western populations. Ultimately, populations retreated to multiple isolated mountainous refugia, shaping the current geographical distribution pattern. These dispersal corridors and montane refugia suggested that the mountains in subtropical China play a crucial role in the conservation of genetic resources and migration of subspecies or related species in this region.

**Keywords:** *Liriodendron chinense*; landscape genetics; genetic structure; refugia; dispersal route

## 1. Introduction

Climate oscillations and topographic changes have significant impacts on the distribution of suitable habitats and migration of species. Since the Cenozoic, the dramatic global climate oscillations of the Quaternary have had a profound effect on the geographical distribution and genetic structure of the plant and animal species in the northern hemisphere [1,2]. During the Quaternary, the global climate underwent several large periodic fluctuations, and repeated glacial and interglacial cycles, which often caused species to retreat to suitable habitat refuges during the glacial periods and then migrate and expand from refuges to other suitable regions after the glacial periods [3,4].

Subtropical China, which is located in the Sino-Japanese Floristic Region of East Asia, harbors diverse temperate flora and was an important glacial refugium for many Tertiary relic species due

to its topographic heterogeneity and complex environments [5,6]. Although geological records indicate that no large-scale ice sheet developed in subtropical China, the climate oscillations of the Quaternary have had a profound influence on the temperate flora of this region [6,7]. The influence of Quaternary climatic fluctuations on the dynamic history of the flora in China's subtropical regions shows diversified patterns, of which there are three: An in situ static pattern, a finite expansion pattern, and an expansion–contraction pattern. The in situ static pattern refers to the expansion and migration of populations in China's subtropical regions before the Quaternary, after which these populations maintained their distribution pattern, which many species continue to exhibit [8,9]. The finite expansion pattern refers to a local contraction of the distribution range of populations during glacial or interglacial periods and local expansion under favorable conditions during glacial or interglacial periods. Numerous classical cases of this pattern have been studied [10–13]. The expansion–contraction pattern refers to populations expanding northward or southward under favorable conditions and contracting to suitable habitats under unfavorable conditions [14,15]. In addition to the impact of climatic fluctuations, the heterogeneous topography also has an important impact on the flora of this region. There are multiple east–west and north–south trending mountains alternating with plains in subtropical China, and these mountains play a significant role in shaping the phylogeographic patterns of species [7]. Extensive phylogeographic studies suggest that the mountains of subtropical China currently play various roles, such as geographic barriers, dispersal corridors, and refugia, in the distribution of different plants [6,10,16–18].

As members of one of the most ancestral angiosperm groups, the species of Magnoliaceae have significant reference value for studies on the origin, historical distribution, dispersal, and phylogeny of angiosperms [19]. The genus *Liriodendron* (Magnoliaceae) includes two extant species, namely, *L. chinense* and *L. tulipifera* [20]. Extensive fossil evidence indicates that *Liriodendron* was still common in the Tertiary throughout the northern hemisphere [21–24], but the genus currently consists of only two sister species with an East Asian–North American disjunct distribution: *L. chinense* and *L. tulipifera*. *L. tulipifera* is prolific throughout the southeastern United States, while *L. chinense* occurs in small, widely scattered populations in subtropical China and northern Vietnam [25]. Due to endangered habitats, intense interspecific competition, low seed viability, and artificial interference, *L. chinense* is restricted to southern areas of the Yangtze River of China and has been listed as a secondary threatened species in China [26,27]. Previous studies on *L. chinense* mainly focused on its biological characteristics, germplasm resources, reproductive biology, wood properties, and hybridization [25,27–30]. Although, some studies have examined the phylogeography of *L. chinense* [31–35], the drivers of its geographical distribution pattern, historical distribution range, and historical dispersal routes are still poorly understood.

Knowledge of the historical geographical distribution, expansion, and migration of species is important for providing insight into their population genetic structure, the historical processes that shaped their current distribution pattern and their probable fate. Therefore, investigating these aspects of *L. chinense* will provide new insight into this species. Although traditional phylogeographic analysis can be used to infer the dynamic history of populations of species, the historical distribution range and routes and direction of expansion and migration of species are still difficult to infer.

In recent years, landscape genetics have been increasing in popularity and developing, and improvements in molecular genetics tools, combined with existing or new statistical methods (Geo-statistics, Bayesian approaches, and maximum likelihood) and powerful computers, have achieved better results by combining genetic and geospatial data [36–38]. The aim of such techniques is to link landscape features and biological microevolutionary processes, such as gene flow, genetic drift, and adaptive evolution [39]. Landscape genetics can resolve population substructure across different geographical regions at low taxonomic levels (interpopulations and subpopulations within species) [40]. The availability of geospatial data (vegetation, climate, paleoclimate, and geology) and the development of predictive modeling approaches [41] have progressed in parallel with these innovations in phylogeographic analysis.

For example, by using species distribution models (SDMs) and combining climate data from different periods (past, present, and future) with geological data, researchers can infer glacial refuges and predict the future distribution range of species [41–43]. Compared with the traditional phylogeographic analysis methods, which uses only molecular data, the integration of phylogeographic analysis and landscape genetics approaches will help us thoroughly understand how patterns of divergence within species, complex population genetic structure, and dispersal routes coincide with current and historical geological and geospatial features [39]. For instance, identifying dispersal corridors and estimating the extent of gene flow in current and historical periods for natural populations could help us uncover the role of population connectivity in divergence [16,39,44,45]. In addition, integrative approaches will provide us with optimal and convenient visualizations of results, which are easier to analyze and understand for researchers. For example, researchers can project a phylogeny onto a 2D or 3D map by integrating genetic data and geospatial data, which helps the investigator understand the relationship between the population genetic structure and geographical distribution pattern of a species [46,47].

Previous studies of the phylogeography of *L. chinense* mainly used genetic data and rarely combined the information with geospatial data [31–33,35]. In this study, we first obtained genetic data (chloroplast DNA (cpDNA), nuclear ribosomal DNA (nrDNA) and nuclear DNA (nDNA)) from 29 populations, climate data (climate layers) from different periods (past, present, and future), and geological data (elevation and slope layer). We then fully integrated these genetic data and geospatial data to reveal the range-wide population genetic structure and infer the historical migration routes and glacial refuges of *L. chinense* in China by combining phylogeographic analysis and landscape genetics approaches (approximate Bayesian computation, SDMs, and GenGIS and Barrier software). This study provides a new perspective with which to understand the subtle influence of historical changes in spatial geographical environments on current geographical distribution patterns of *L. chinense*.

## 2. Materials and Methods

### 2.1. Populations and Sampling

Twenty-nine natural populations were sampled throughout the distribution range of *L. chinense*. We obtained cpDNA (*psb*A-*trn*H and *trn*T-*trn*L) and nrDNA (internal transcribed spacer (ITS)) from 23 population samples (Table 1, Figure 1a). A total of 111 individuals were utilized to obtain cpDNA and nrDNA data. Although cpDNA and nrDNA are favored for phylogeographic analyses, especially in plants [16,48], a single locus or two loci may not accurately reflect population boundaries and phylogeography. Therefore, six additional nuclear gene sequences were obtained from 23 populations of samples to expand our data set (Table 1, Figure 1b). Given that *L. chinense* is an endangered species, the size of its natural populations is generally small: Most populations include only one to ten or ten to twenty individuals. These natural populations vary in size and are scattered in subtropical China mostly at altitudes around 700–1900 m [26,27]. In this study, the spatial distance between each individual is 50 × 50 m. If there were less than five samples in the populations, we collected all the individuals in this population. In addition, DNA of most trees undergoes genetic recombination, and the genetic diversity of species is usually studied by calculating the frequency of traditional genetic markers (e.g., diversity arrays technology (DArT), simple sequence repeats (SSRs), and restriction site associated DNA sequencing (RAD-seq)) in the populations because these types of markers are affected by genetic recombination. With such methods, more samples of each population are needed to achieve statistical significance. Relatively speaking, functional genes are less affected by genetic recombination, and genetic variation is stable between two adjacent generations. Only a small number of samples are needed for each population to represent the genetic variation in the populations [39,49]. In this study, we collected one to four individuals from each population and obtained six low-copy functional gene sequences from 23 populations. The samples of leaves or buds we collected were all from adults. After collection, each plant material sample was stored in a plastic bag with a 100 g silica desiccant pack at room temperature pending DNA extraction.

**Table 1.** The sample size, location, abbreviated name, altitude, and geographical coordinates of the sampled *L. chinense* populations.

| Abbreviation | Location | Longitude | Latitude | Altitude (m) | Sample Size |
|---|---|---|---|---|---|
| AJ | Anji, Zhejiang, CHN | 119.43° E | 30.4° N | 935–1000 | 5 |
| SY | Songyang, Zhejiang, CHN | 119.6° E | 28.5° N | 138 | 5 |
| SC | Suichang, Zhejiang, CHN | 118.83° E | 28.41° N | 880–1410 | 5 |
| JX | Jixi, Anhui, CHN | 118.83° E | 30.12° N | 750–1190 | 5 |
| HS | Huangshan, Anhui, CHN | 116.1° E | 30.17° N | 1250 | 5 |
| LS | Lushan, Jiangxi, CHN | 116° E | 29.53° N | 1167 | 5 |
| FJWYS | Wuyishan, Fujian, CHN | 117.76° E | 27.84° N | 1700 | 5 |
| JXWYS | Wuyishan, Jiangxi, CHN | 117.8° E | 27.92° N | 873 | 5 |
| XN | Xianning, Hubei, CHN | 114.2° E | 29.8° N | 68 | 5 |
| EX | Exi, Hubei, CHN | 109° E | 30.3° N | 1180 | 5 |
| SN | Suining, Hunan, CHN | 110.2° E | 26.33° N | 1500 | 5 |
| SZ | Sangzhi, Hunan, CHN | 110.2° E | 29.15° N | 407 | 5 |
| JJ | Jiaojiang, Guangxi, CHN | 110.84° E | 25.56° N | 496 | 5 |
| MES | Maoershan, Guangxi, CHN | 110.4° E | 25.87° N | 1100–1200 | 5 |
| HP | Huaping, Guangxi, CHN | 110.37° E | 25.88° N | 1280 | 5 |
| YC | Yachang, Guizhou, CHN | 106.22° E | 24.9° N | 594 | 1 |
| ST | Songtao, Guizhou, CHN | 109.32° E | 28.16° N | 903–937 | 5 |
| XS | Xishui, Guizhou, CHN | 105.89° E | 28.24° N | 1100–1250 | 5 |
| LP | Liping, Guizhou, CHN | 109.19° E | 26.34° N | 421 | 5 |
| XY | Xuyong, Sichuan, CHN | 105.5° E | 28.2° N | 800 | 5 |
| YY | Youyang, Sichuan, CHN | 108.8° E | 28.82° N | 890 | 5 |
| JP | Jinping, Yunnan, CHN | 103.23° E | 22.78° N | 1230 | 5 |
| MLP | Malipo, Yunnan, CHN | 104.47° E | 23.3° N | 1420–1480 | 5 |
| JDZ | Jingdezhen, Jiangxi, CHN | 117.66° E | 29.55° N | 530 | 1 |
| LY | Leye, Guangxi, CHN | 106.3° E | 24.84° N | 1044 | 3 |
| YJ | Yinjiang, Guizhou, CHN | 108.61° E | 27.89° N | 1560 | 3–4 |
| XC | Xichou, Yunnan, CHN | 104.47° E | 23.3° N | 1480 | 4 |
| MG | Maguan, Yunan, CHN | 104.19° E | 23.02° N | 1420 | 4 |
| YUY | Yuanyang, Yunnan, CHN | 103.07° E | 23.03° N | 1540–1600 | 3 |

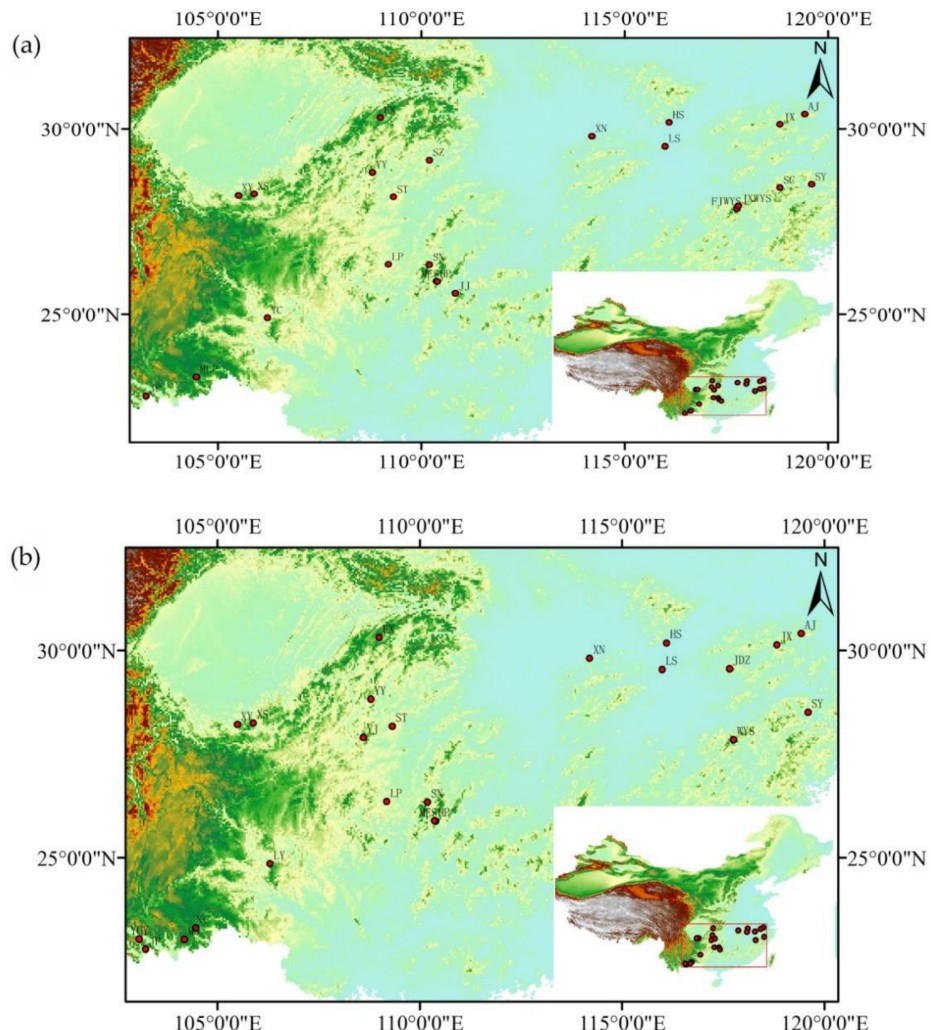

**Figure 1.** Sample location distribution of *L. chinense*. (**a**) cpDNA and nrDNA from 23 populations. (**b**) Six nuclear genes from 23 populations.

*2.2. DNA Extraction and Sequence Data Acquisition*

The genomic DNA was extracted by using a modified cetyl trimethylammonium bromide (CTAB) method [50]. To obtain suitable chloroplast fragments for *L. chinense*, the angiosperm chloroplast genome fragments from previous reports were screened, and two chloroplast fragments, namely, *psbA-trn*H and *trn*T-*trn*L, were selected. The primers of these two sequences were obtained from the literature [51,52]. Primers of the ITS were obtained from a previous report [53]. Primers of six nuclear genes (*LcDHN-like*, *LcDHN-like1*, *LcDHN-like2*, *LtNCED1*, *LtNCED3*, and *Ltosmotin-like*) were obtained from the expressed sequence tag (EST) fragments of the corresponding genes in the *L. chinense* transcriptome database [54] and the *L. tulipifera* EST database (http://ancangio.uga.edu/) [55]. The specific primer sequences of the corresponding genes are shown in Table 2. PCR was performed in a 50-µL reaction system according to the following cycling protocol: Initial denaturation at 94 °C for 3 min; 35 cycles of 94 °C for 30 s, the PCR annealing temperature (T °C) for 30 s, and 72 °C for 1 min; followed by a final extension at 72 °C for 10 min. PCR products were segregated on a 1.5% agarose gel and purified using a DNA gel extraction kit (Tiangen Biotech, Beijing, China). The purified PCR products were cloned into pEASY-T1 cloning vector and transferred into Trans5α chemically competent cells for positive monoclonal selection (Tiangen Biotech, Beijing, China). One to four positive monoclones of each individual were selected for sequencing.

**Table 2.** All locus primers used in this study.

| Locus | Primer Pairs (5′–3′) | Reference |
|---|---|---|
| *psb*A-*trn*H | F: CGCATGGTGGATTCACAATC | [51] |
| | R: AGACCTAGCTGCTATCGAAG | |
| *trn*T-*trn*L | F: CATTACAAATGCGATGCTCT | [52] |
| | R: TCTACCGATTTCGCCATATC | |
| ITS | F: TACCGATTGAATGATCCGGTGAAG | [53] |
| | R: CGCCGTTACTAGGGGAATCCTTGT | |
| *LcDHN-like* | F: GTAGTTGATTTTGAGCCGTT | Newly designed |
| | R: CACACATCCTACTTGTGACCT | |
| *LcDHN-like1* | F: AAAAGCAAAAGCTCTTCG | Newly designed |
| | R: CATCAATCAAAAGGACACAAA | |
| *LcDHN-like2* | F: ATGGGGAAGAAGGAAGAAAAG | Newly designed |
| | R: TCAGTGGTTGGCAGACTC | |
| *LtNCED1* | F: ATTCTTCCCATTCTACACT | Newly designed |
| | R: TCTCCCCTCCTCTAACCAA | |
| *LtNCED3* | F: ATGGCGACTGCAAGTAGTA | Newly designed |
| | R: TTAGACCTGGCTCACCAG | |
| *Ltosmotin-like* | F: ATGGGGAACGCTCCAAC | Newly designed |
| | R: TTAGTGGCAAAAGATAACCTTC | |

## 2.3. Data Analysis

### 2.3.1. Genetic Diversity and Test of Population Expansion

DNA sequences were aligned using ClustalX 2.10 (European Molecular Biology Laboratory, Heidelberg, Germany) [56]. The genetic diversity parameters of populations were estimated for chloroplast and nuclear loci using DnaSP v5 DnaSP v5 (Universitat de Barcelona, Barcelona, Spain) [57]. The number of segregating sites (S), the nucleotide diversity (π), Watterson's parameter (θw), and the haplotype diversity (Hd) were calculated for all loci. Three neutrality tests (Tajima's D, Fu and Li's F *, and Fu and Li's D *) [58,59] were conducted in DnaSP v5 to detect whether populations of *L. chinense* deviated from the neutral evolutionary model and to detect historical population expansion events at different loci [57].

### 2.3.2. Population Genetic Structure and Barrier Analysis

In this study, GenGIS (Faculty of Computer Science, Dalhousie University, Halifax, Canada) and Barrier software were used to reveal the genetic structure and genetic barriers of the *L. chinense* populations, respectively. First, MEGA v7.0 (Département Hommes, Natures, Sociétés, Human Population Genetics Group, Paris, France) [60] software was used to generate neighbor-joining (NJ) trees of cpDNA, nrDNA, and nDNA, and the NJ trees were then overlaid on the topographic map by GenGIS v2.5.3 (Faculty of Computer Science, Dalhousie University, Halifax, Canada) [47]. Areas of abrupt genetic differentiation over relatively short geographic distances can indicate genetic barriers in a species range where distinct phylogeographic lineages converge [61]. Based on Fst matrices of multiple loci and single loci, Barrier v2.2 (Département Hommes, Natures, Sociétés, Human Population Genetics Group, Paris, France) was used to identify genetic barriers and population boundaries [62].

### 2.3.3. Population Connectivity: Visualizing Putative Dispersal Routes

DNA of most trees undergoes genetic recombination. It is difficult to obtain haplotype data from traditional genetic markers (e.g., DArT, SSRs, and RAD-seq), which are affected by genetic recombination. Relatively speaking, functional genes which are less affected by genetic recombination are more advantageous to use a single gene locus marker. Therefore, in this study, we combined the haplotype data with geospatial data to find the dispersal corridors of the species. The median-joining

method in PopART software was used [63] to obtain haplotype networks of all loci. Using MaxEnt 3.3.3 k software (AT and T Labs-Research, Florham Park, NJ, USA) [41], an SDM of *L. chinense* was generated. Afterwards, SDMtoolbox (Department of Zoology, Cooperative Wildlife Research Laboratory, Southern Illinois University at Carbondale, Carbondale, IL, USA) [64] was implemented to convert the model to a friction surface by inverting the SDM. The areas of high suitability were converted into areas of low dispersal cost. Finally, friction surface layers and shared haplotype networks were combined to generate the least-cost paths (LCPs) and dispersal network map among all shared haplotypes with varying sampling localities by the "Calculate Least-Cost Corridors" tool of SDMtoolbox [64].

Based on the chloroplast and nuclear haplotypes shared among regions, gene flow may have been substantial [39]. Therefore, MIGRATE 3.6.11 (Department of Scientific Computing, Florida State University, Tallahassee, FL, USA) [65] software was used to combine six nuclear gene sequences in order to determine the direction and extent of gene flow among populations. Bayesian inference was utilized to calculate the probabilities of explicit population models by coalescence theory [66]. For each model, four parallel static chains were executed at temperatures of 1.0, 1.5, 3.0, and $10^6$, with 100,000 recorded steps and three replicate runs. Using MIGRATE software, marginal likelihood (ML) values of the Bezier approximation score were calculated. We directly determined which model was more likely based on the ML values. The higher the ML value was, the higher the probability of the model [67].

### 2.3.4. Species Distribution Modeling

The geographical distribution ranges of *L. chinense* in China were inferred for the current (1970–2000) [68], last interglacial (LIG, ~120,000–140,000 years BP) [69], last glacial maximum (LGM, ~22,000 years ago, from the CCSM4 model), mid-Holocene (MH, ~6000 years ago, from the CCSM4 model) and future (2060–2080, from the CCSM4 model) time periods [70]. Occurrence records of *L. chinense* were collected from the Chinese Virtual Herbarium (CVH, http://www.cvh.ac.cn/), the Global Biodiversity Information Facility (GBIF, http://www.gbif.org/), and field sampling locations. To reduce error, repetitive geographic coordinate information was deleted, and 145 geographic coordinate locations of *L. chinense* were finally obtained (Table S1).

The environmental variables consisted of 19 bioclimatic variables and 2 topographical variables (Table S2). All layers of environmental variables had a 2.5-arc minute resolution. In SDMs, multicollinearity of variables may violate statistical assumptions and lead to overfitting [71]. Therefore, the "Band Collection Statistic" tool in ArcGIS 10.2 was used to calculate the Pearson correlation coefficients between pairs of 21 environmental variables, and the variables with a correlation coefficient ($|r| \geq 0.8$) were removed. Ultimately, 12 environmental variables were adopted in this study (Table 3).

**Table 3.** Environmental variables used in the MaxEnt model and their percent contribution.

| Variables | Description | Unit | Contribution (%) |
|---|---|---|---|
| Bio2 | Mean Diurnal Range (Mean of monthly (max temp–min temp)) | °C | 2.3 |
| Bio3 | Isothermality (Bio2/Bio7) (* 100) | | 3.5 |
| Bio4 | Temperature Seasonality (standard deviation * 100) | °C | 0.2 |
| Bio8 | Mean Temperature of Wettest Quarter | °C | 3.3 |
| Bio9 | Mean Temperature of Driest Quarter | °C | 7.3 |
| Bio11 | Mean Temperature of Coldest Quarter | °C | 0.3 |
| Bio13 | Precipitation of Wettest Month | mm | 1.6 |
| Bio15 | Precipitation Seasonality (Coefficient of Variation) | | 0.8 |
| Bio16 | Precipitation of Wettest Quarter | mm | 6.1 |
| Bio17 | Precipitation of Driest Quarter | mm | 65.3 |
| Elevation | | m | 2.3 |
| Slope | | ° | 6.8 |

MaxEnt 3.3.3k was used to simulate the potential geographically suitable range of *L. chinense* during the five periods [41]. Seventy-five percent of the sites were randomly selected for model construction, and the remaining 25% sites were used to test and validate the model, with 10 repeated runs for each simulation. In addition, the jackknife procedure was performed to ascertain the environmental variables that made the greatest contribution to the distribution range of *L. chinense* [72]. After the simulation, the receiver operating characteristic (ROC) and area under the ROC curve in the simulation results were used to evaluate the reliability of the models [73]. An area under the curve (AUC) of >0.9 indicates that the simulation result is accurate, and an AUC > 0.8 indicates that the simulation result can be adopted [41].

## 3. Results

### 3.1. Analysis of Genetic Diversity and Test for Population Expansion

In this study, genetic polymorphisms at nine loci in eastern and western populations of *L. chinense* were analyzed (Table 4). The nucleotide diversity ($\pi$) of four loci (*trnT-trn*L, nrDNA, *LcDHN-like1*, and *LcDHN-like2*) in the western populations was higher than that in the eastern populations, while the nucleotide diversity ($\pi$) of the remaining five loci (*psbA-trn*H, *LcDHN-like*, *LtNCED1*, *LtNCED3*, and *Ltosmotin-like*) in the western populations was lower than that in the eastern populations. A *t* test revealed no significant difference in nucleotide diversity between the eastern and western populations (0.00393 ± 0.00315 vs. 0.00393 ± 0.00243, respectively, *p* > 0.05, *df* = 8). Watterson's parameter ($\theta_w$) of five loci (*trnT-trn*L, ITS, *LcDHN-like2*, *LtNCED3*, and *Ltosmotin-like*) in the western populations was higher than that in the eastern populations, while Watterson's parameter ($\theta_w$) of the remaining four loci (*psbA-trn*H, *LcDHN-like*, *LcDHN-like1*, and *LtNCED1*) in the western populations was lower than that in the eastern populations. A *t* test revealed no significant difference in $\theta_w$ between the eastern and western populations (0.00575 ± 0.00343 vs. 0.00654 ± 0.00383, respectively, *p* > 0.05, *df* = 8). The haplotype diversity (Hd) of six loci (*psbA-trn*H, *trn*T-*trn*L, ITS, *LcDHN-like*, *LcDHN-like2*, and *LtNCED1*) in the western populations was higher than that in the eastern populations, while the Hd of the remaining three loci (*LcDHN-like1*, *LtNCED3*, and *Ltosmotin-like*) in the western populations was lower than that in the eastern populations. A *t* test revealed no significant difference in Hd between the eastern and western populations (0.726 ± 0.317 vs. 0.797 ± 0.225, respectively, *p* > 0.05, *df* = 8). In addition, genetic polymorphisms were revealed by cpDNA and nDNA in natural populations of *L. chinense* (Table 5). The nucleotide diversity ($\pi$), Watterson's parameter ($\theta$w), and the haplotype diversity (Hd) of cpDNA in the western populations was higher than that in the eastern populations, possibly because we collected more samples from the western populations than from eastern populations. Six nuclear genes and the nrITS locus from 13 populations were analyzed (Table 5), and approximate genetic polymorphisms were detected between the eastern and western populations.

Tajima's D, Fu and Li's F * and Fu and Li's D * were significantly negative for the nrDNA ITS in the eastern populations, western populations and whole range of *L. chinense* (Table 6). Tajima's D, Fu and Li's F * and Fu and Li's D * were significantly negative for *LcDHN-like2* for the whole range of *L. chinense*. For the western populations, Fu and Li's F * and Fu and Li's D * were significantly negative at *LcDHN-like2*. For the whole range of *L. chinense*, Fu and Li's F * and Fu and Li's D * were significantly negative for *Ltosmotin-like*. Therefore, the eastern and western populations might have undergone population expansion or migration events during their evolutionary histories. The other six loci did not deviate from the neutral evolutionary model in the eastern populations, western populations, or whole range of *L. chinense*.

**Table 4.** Summary statistics of sequence diversity of *L. chinense*.

| Locus | Population | $N$ | $L$ | $S$ | $\pi$ | $\theta_w$ | $H$ | $Hd$ |
|---|---|---|---|---|---|---|---|---|
| *psb*A-*trn*H | East | 45 | 505 | 19 | 0.00652 | 0.00889 | 9 | 0.543 |
| | West | 65 | 505 | 19 | 0.00625 | 0.00771 | 10 | 0.547 |
| | Whole range | 110 | 505 | 25 | 0.00802 | 0.00970 | 16 | 0.660 |
| *trn*T-*trn*L | East | 45 | 868 | 0 | 0.00000 | 0.00000 | 1 | 0.000 |
| | West | 65 | 868 | 4 | 0.00042 | 0.00101 | 5 | 0.328 |
| | Whole range | 110 | 868 | 4 | 0.00025 | 0.00091 | 5 | 0.204 |
| nrDNA | East | 50 | 862 | 25 | 0.00146 | 0.00694 | 16 | 0.567 |
| | West | 65 | 862 | 43 | 0.00364 | 0.01147 | 30 | 0.865 |
| | Whole range | 115 | 862 | 60 | 0.00196 | 0.01453 | 38 | 0.596 |
| *LcDHN-like* | East | 48 | 1082 | 54 | 0.01047 | 0.01136 | 21 | 0.940 |
| | West | 90 | 1082 | 69 | 0.00887 | 0.01293 | 43 | 0.976 |
| | Whole range | 138 | 1082 | 92 | 0.01005 | 0.01590 | 61 | 0.981 |
| *LcDHN-like1* | East | 38 | 3536 | 65 | 0.00371 | 0.00532 | 28 | 0.976 |
| | West | 58 | 3536 | 70 | 0.00414 | 0.00515 | 29 | 0.956 |
| | Whole range | 96 | 3536 | 101 | 0.00421 | 0.00677 | 54 | 0.979 |
| *LcDHN-like2* | East | 36 | 1440 | 22 | 0.00173 | 0.00376 | 14 | 0.814 |
| | West | 60 | 1440 | 29 | 0.00278 | 0.00441 | 22 | 0.892 |
| | Whole range | 96 | 1440 | 44 | 0.00259 | 0.00609 | 33 | 0.915 |
| *LtNCED1* | East | 38 | 1884 | 27 | 0.00237 | 0.00323 | 20 | 0.939 |
| | West | 58 | 1884 | 24 | 0.00197 | 0.00260 | 29 | 0.954 |
| | Whole range | 96 | 1884 | 41 | 0.00232 | 0.00401 | 48 | 0.973 |
| *LtNCED3* | East | 40 | 1884 | 36 | 0.00388 | 0.00452 | 24 | 0.971 |
| | West | 60 | 1884 | 42 | 0.00296 | 0.00480 | 35 | 0.967 |
| | Whole range | 100 | 1884 | 60 | 0.00403 | 0.00618 | 57 | 0.983 |
| *Ltosmotin-like* | East | 40 | 774 | 29 | 0.00469 | 0.00881 | 11 | 0.800 |
| | West | 60 | 774 | 33 | 0.00457 | 0.00930 | 15 | 0.737 |
| | Whole range | 100 | 774 | 40 | 0.00463 | 0.01015 | 22 | 0.769 |

$N$: The number of sequences; $L$: The number of sites in aligned sequences; $S$: The number of segregating sites; $\pi$: Nucleotide diversity; $\theta_w$: Watterson's parameter; $H$: The number of haplotypes; $Hd$: Haplotype diversity.

**Table 5.** Genetic diversity revealed by chloroplast DNA (cpDNA) and nuclear DNA (nDNA) in natural populations of *L. chinense*.

| Pop (cpDNA) | $N$ | $\pi$ | $\theta_w$ | $H$ | $Hd$ | Pop (nDNA) | $N$ | $\pi$ | $\theta_w$ |
|---|---|---|---|---|---|---|---|---|---|
| AJ | 5 | 0.00045 | 0.00036 | 2 | 0.600 | AJ | 6 | 0.00312 | 0.00316 |
| HS | 5 | 0.00090 | 0.00107 | 3 | 0.700 | SY | 6 | 0.00222 | 0.00233 |
| JX | 5 | 0.00134 | 0.00107 | 2 | 0.600 | HS | 4 | 0.00331 | 0.00305 |
| JXWYS | 5 | 0.00223 | 0.00250 | 3 | 0.800 | LS | 6 | 0.00319 | 0.00324 |
| LS | 5 | 0.00030 | 0.00036 | 2 | 0.400 | FJWYS | 4 | 0.00325 | 0.00349 |
| SC | 4 | 0.00050 | 0.00041 | 2 | 0.667 | XN | 4 | 0.00315 | 0.00309 |
| SY | 5 | 0.00045 | 0.00036 | 2 | 0.600 | EX | 6 | 0.00358 | 0.00360 |
| FJWYS | 5 | 0 | 0 | 1 | 0 | SN | 6 | 0.00328 | 0.00326 |
| XN | 5 | 0.00492 | 0.00465 | 4 | 0.900 | MES | 6 | 0.00281 | 0.00265 |
| EX | 4 | 0.00336 | 0.00367 | 4 | 1.000 | ST | 2 | 0.00137 | 0.00137 |
| HP | 5 | 0.00060 | 0.00071 | 2 | 0.400 | LP | 6 | 0.00235 | 0.00222 |
| SN | 5 | 0.00313 | 0.00251 | 3 | 0.800 | XY | 4 | 0.00108 | 0.00110 |
| JJ | 5 | 0.00268 | 0.00214 | 2 | 0.600 | YY | 6 | 0.00270 | 0.00272 |
| JP | 5 | 0.00030 | 0.00036 | 2 | 0.400 | JX | | | |
| LP | 5 | 0.00268 | 0.00321 | 3 | 0.700 | JDZ | | | |
| MES | 5 | 0.00090 | 0.00109 | 3 | 0.700 | YJ | | | |
| MLP | 4 | 0.00037 | 0.00041 | 2 | 0.500 | HP | | | |
| ST | 5 | 0.00231 | 0.00287 | 4 | 0.900 | XS | | | |
| SZ | 5 | 0.00030 | 0.00036 | 1 | 0.400 | LY | | | |
| YY | 5 | 0.00340 | 0.00287 | 3 | 0.700 | XC | | | |
| XS | 3 | 0.00372 | 0.00397 | 3 | 1 | JP | | | |
| XY | 4 | 0.00099 | 0.00081 | 2 | 0.667 | MG | | | |
| YC | 1 | | | | | YUY | | | |
| East | 44 | 0.00243 | 0.00327 | 9 | 0.553 | East | 30 | 0.00376 | 0.00504 |
| West | 61 | 0.00258 | 0.00356 | 14 | 0.665 | West | 36 | 0.00365 | 0.00522 |
| Whole | 105 | 0.00313 | 0.00421 | 20 | 0.719 | Whole | 66 | 0.00406 | 0.00685 |

**Table 6.** Neutrality test values of all loci based on nucleotide variation.

| Locus | Population | Tajima's D | Fu and Li's D * | Fu and Li's F * |
|---|---|---|---|---|
| *psb*A-*trn*H | East | −0.85991 | −0.20734 | −0.50775 |
| | West | −0.70881 | −1.21672 | −1.23169 |
| | Whole range | −0.60507 | −2.27784 | −1.94985 |
| *trn*T-*trn*L | East | | | |
| | West | −1.25638 | −1.30785 | −1.51234 |
| | Whole range | −1.40697 | −1.50383 | −1.73357 |
| nrDNA | East | −2.58119 *** | −5.68498 ** | −5.45963 ** |
| | West | −2.31027 ** | −4.69552 ** | −4.53354 ** |
| | Whole range | −2.76256 *** | −7.68778 ** | −6.76095 ** |
| *LcDHN-like* | East | −0.09932 | 0.48782 | 0.30949 |
| | West | −0.94272 | 0.06387 | −0.44434 |
| | Whole range | −1.10330 | −0.39386 | −0.86741 |
| *LcDHN-like1* | East | −1.05348 | −0.48634 | −0.83987 |
| | West | −0.44559 | 0.99637 | 0.49643 |
| | Whole range | −1.21302 | −0.22537 | −0.80095 |
| *LcDHN-like2* | East | −1.69227 | −0.02854 | −0.72973 |
| | West | −1.37388 | −2.62366 * | −2.55638 * |
| | Whole range | −1.85038 * | −2.74296 * | −2.83902 * |
| *LtNCED1* | East | −0.58354 | 0.59739 | 0.20543 |
| | West | −0.97734 | −1.04917 | −1.22219 |
| | Whole range | −1.29488 | −0.76723 | −1.18870 |
| *LtNCED3* | East | −0.44472 | −0.81597 | −0.80943 |
| | West | −1.20350 | −0.34101 | −0.81769 |
| | Whole range | −1.03756 | −1.81555 | −1.77416 |
| *Ltosmotin-like* | East | −1.34011 | −1.20867 | −1.50054 |
| | West | −1.59654 | −1.82745 | −2.08065 |
| | Whole range | −1.67898 | −3.75720** | −3.45120** |

\* significant; $p < 0.05$ ** significant; $p < 0.02$ *** significant; $p < 0.01$.

### 3.2. Population Genetic Structure

Based on cpDNA (*psbA-trn*H and *trn*T-*trn*L) and nrDNA from 23 populations and five nuclear genes (*LcDHN-like1*, *LcDHN-like2*, *LtNCED1*, *LtNCED3*, and *Ltosmotin-like*) from 16 populations, NJ trees were overlaid on topographic maps (Figure 2a–c). The NJ tree of two tandem chloroplast fragments (*psbA-trn*H and *trn*T-*trn*L) clustered most eastern populations into a clade separated from the western populations, while minority populations (SZ and FJWYS) arose from admixture between eastern and western populations (Figure 2a). The NJ tree of nrDNA also showed genetic differentiation between the eastern and western populations, but minority populations (SC and SN) arose from admixture (Figure 2b). The NJ tree of five tandem nuclear genes (*LcDHN-like1*, *LcDHN-like2*, *LtNCED1*, *LtNCED3*, and *Ltosmotin-like*) from 16 populations divided the populations into two clades. One clade included nine western populations (EX, SN, ST, YY, LY, MES, LP, XY, and YUY) and an eastern population (XN); the other clade included five eastern populations (AJ, HS, SY, WYS, and LS) and a western population (YJ) (Figure 2c). Moreover, the XN and YJ populations arose from admixture between the eastern and western populations.

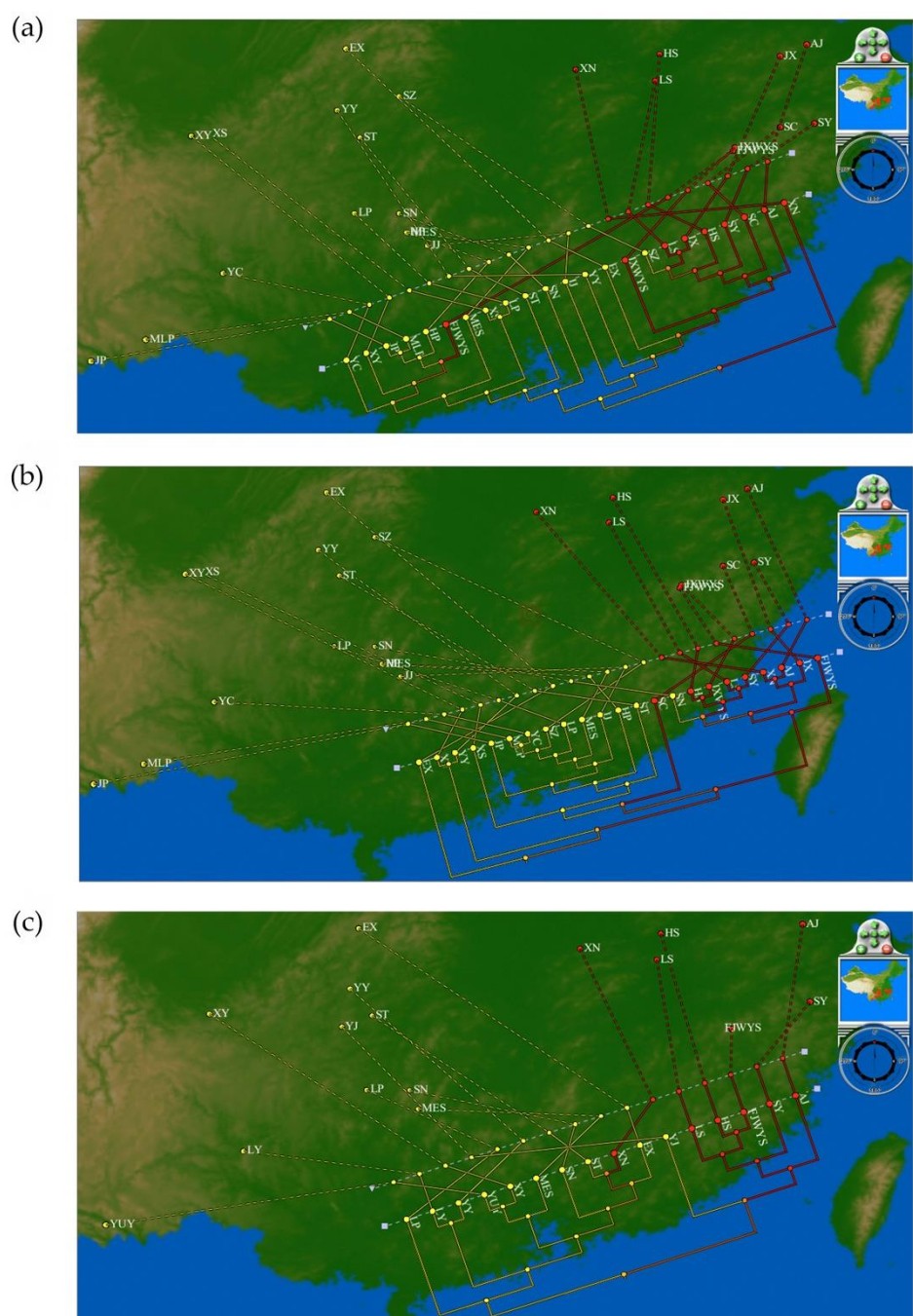

**Figure 2.** Neighbor-joining (NJ) trees of *L. chinense* populations. (**a**) NJ trees of *L. chinense* populations based on cpDNA. (**b**) NJ trees of *L. chinense* populations based on nuclear ribosomal DNA (nrDNA). (**c**) NJ trees of *L. chinense* populations based on five nuclear genes. Red dots represent sample locations of eastern populations; yellow dots represent sample locations of western populations. The colors of NJ tree branches indicate sample locations.

Results from the NJ tree revealed genetic differentiation between the eastern and western populations. Therefore, Barrier v2.2 software (Département Hommes, Natures, Sociétés, Human Population Genetics Group, Paris, France) was used to detect genetic barriers between these eastern and western populations of *L. chinense*. The results from the barrier analysis were consistent with the results of the NJ tree. Based on Fst matrices of cpDNA, two barriers were identified among 23 sample populations (Figure 3a). The first barrier separated one eastern population (XN) from the remaining eastern populations, and the second barrier separated the XN population from the western populations. In addition, based on Fst matrices of five tandem nuclear genes, we also identified two barriers among the 16 sample populations (Figure 3b). The first barriers separated one western population (YJ) from the remaining western populations, and the second barrier separated the XN population from the remaining eastern populations.

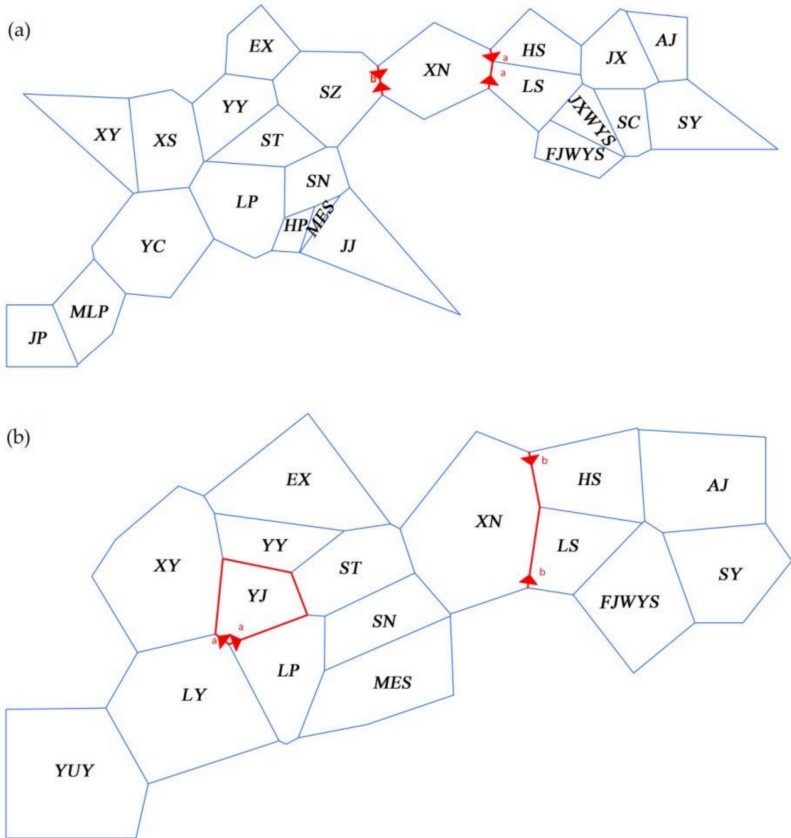

**Figure 3.** The first two genetic barriers in *L. chinense* projected by Barrier 2.2. (**a**) The first two genetic barriers in *L. chinense* based on cpDNA. (**b**) The first two genetic barriers in *L. chinense* based on five nuclear genes. The red lines with two opposite arrows indicate the barriers (**a,b**). The black letters represent the abbreviated names of sample locations.

*3.3. Visualizing Dispersal Corridors and Gene Flow Direction*

Although we detected genetic divergence between the eastern and western populations, genetic admixture events were still detected in minority populations (Figure 2a–c). Following the concept of landscape genetics, we combined genetic data and SDMs and explored whether there were dispersal corridors between the eastern and western populations of *L. chinense*.

The dispersal networks of cpDNA, nrDNA, and two nuclear genes (*LcDHN-like2* and *Ltosmotin-like*) indicated two significant dispersal corridors between the eastern and western populations of *L. chinense* (Figure 4a–d). A dispersal corridor connected northeastern and northwestern populations of *L. chinense*. Another dispersal corridor connected southeastern and southwestern populations of *L. chinense*. The distribution of haplotypes from four loci showed that 1–3 haplotypes were shared by many eastern and

western populations, especially for cpDNA, nrDNA, and *Ltosmotin-like* (Figure 5a–d). Another dispersal corridor connecting southeastern and southwestern populations was not obvious for *LcDHN-like*2, which was consistent with the distribution of its haplotypes (Figures 4b and 5b). In addition, there was a dispersal corridor connecting the north and south within eastern and western populations. The distribution area of *L. chinense* formed an approximately closed circular dispersal corridor. The dispersal networks of two nuclear genes (*LcDHN-like* and *LcDHN-like1*) showed no dispersal corridor between the eastern and western populations except the XN population, and the eastern and western populations were isolated (Figure 4e,g). The distribution of haplotypes of these two loci revealed a few haplotypes shared by several eastern and western populations (Figure 5e,g). The haplotype networks of the two loci showed that the eastern populations were distinct from the western populations (Figure S1). The dispersal network of the nuclear gene *LtNCED1* detected dispersal corridors between some adjacent sampling sites (Figure 4f). The distribution of haplotypes of *LtNCED1* showed a limited number of haplotypes shared in several nearby populations (Figure 5f). The haplotype network exhibited a complex structure at this locus (Figure S1). The dispersal network of the nuclear gene *LtNCED3* showed a single dispersal corridor between the eastern and western populations (Figure 4h). The distribution of haplotypes at *LtNCED3* revealed rare haplotypes shared between a few eastern and western populations (Figure 5h). The haplotype network of *LtNCED3* showed that the eastern populations were obviously separated from the western populations (Figure S1). In addition, we found many endemic haplotypes in the eastern and western populations, especially at *LcDHN-like*, *LcDHN-like1*, *LtNCED1*, and *LtNCED3* (Figure 5e–h).

Because dispersal corridors were detected between the eastern and western populations, we further tested whether there was potential historical gene flow between these eastern and western populations. The sampling sites were divided into four regions: Northwestern (NW), northeastern (NE), southwestern (SW), and southeastern (SE) populations (Figure 6c–d). Based on sequences of six nuclear genes, the direction and extent of gene flow between six population pairs were determined. Unidirectional gene flow was detected in all population pairs (NW→NE, SW→NE, SE→SW, SE→NW, SE→NE, and NW→SW) (Table 7). Abundant gene flow was found for most population pairs (NW→NE, SW→NE, SE→NW, SE→NE, and NW→SW), and only a small amount of gene flow was detected in one population pair (SE→SW) (Table 8). In addition, there was a larger amount of gene flow from the western population to the eastern population than in the opposite direction (Table 8), which may be because the western populations have a larger effective population size (Figure 6c). Furthermore, the amount of gene flow within eastern and western populations was larger than that between eastern and western populations (Table 8), which may be due to the longer geographical distance between eastern and western populations than between populations in each group (Figure 6d).

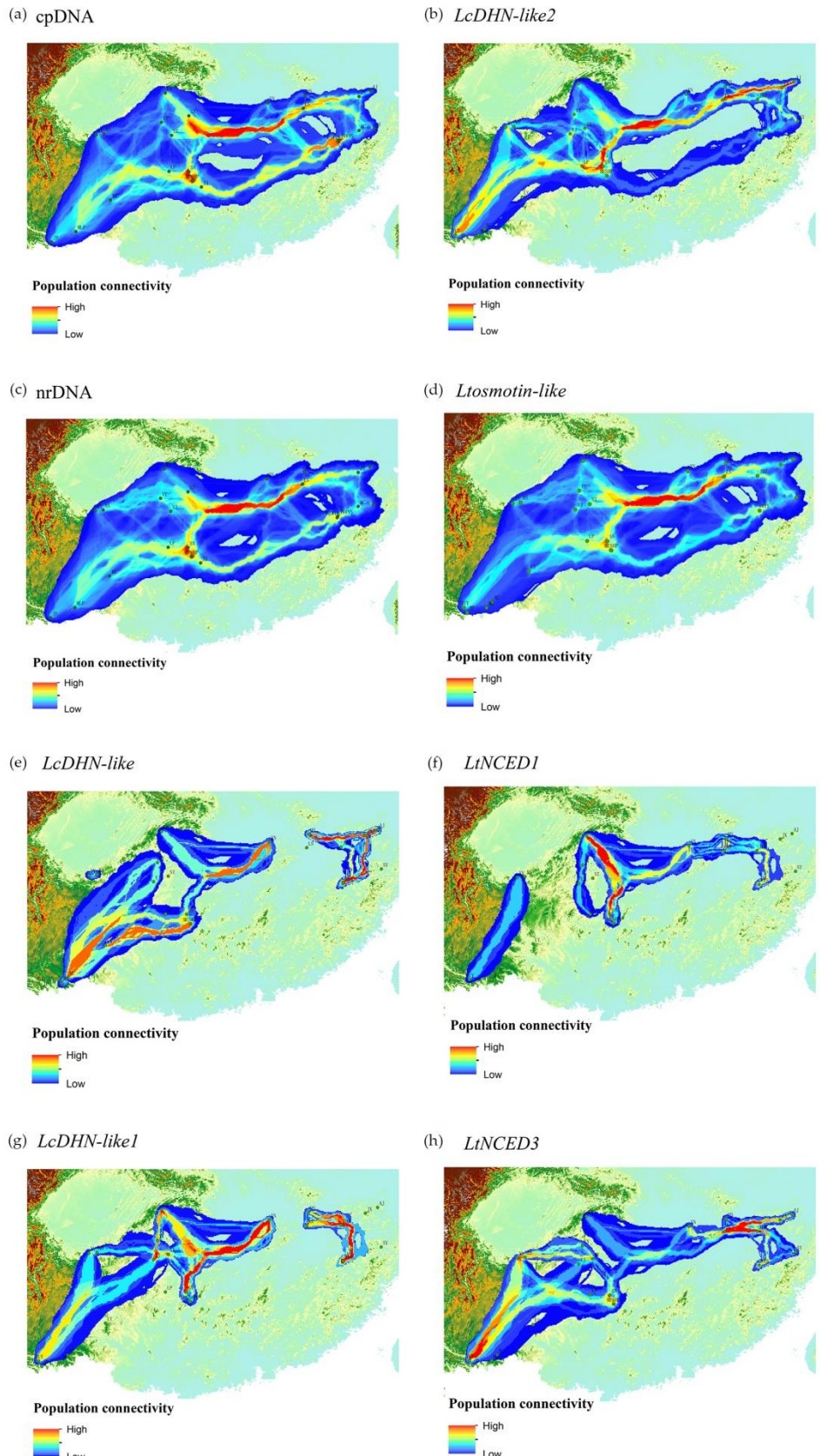

**Figure 4.** Construction of dispersal networks for *L. chinense* based on cpDNA, nrDNA, and six nuclear genes sequences (**a**–**h**). Warmer color depicts higher population connectivity.

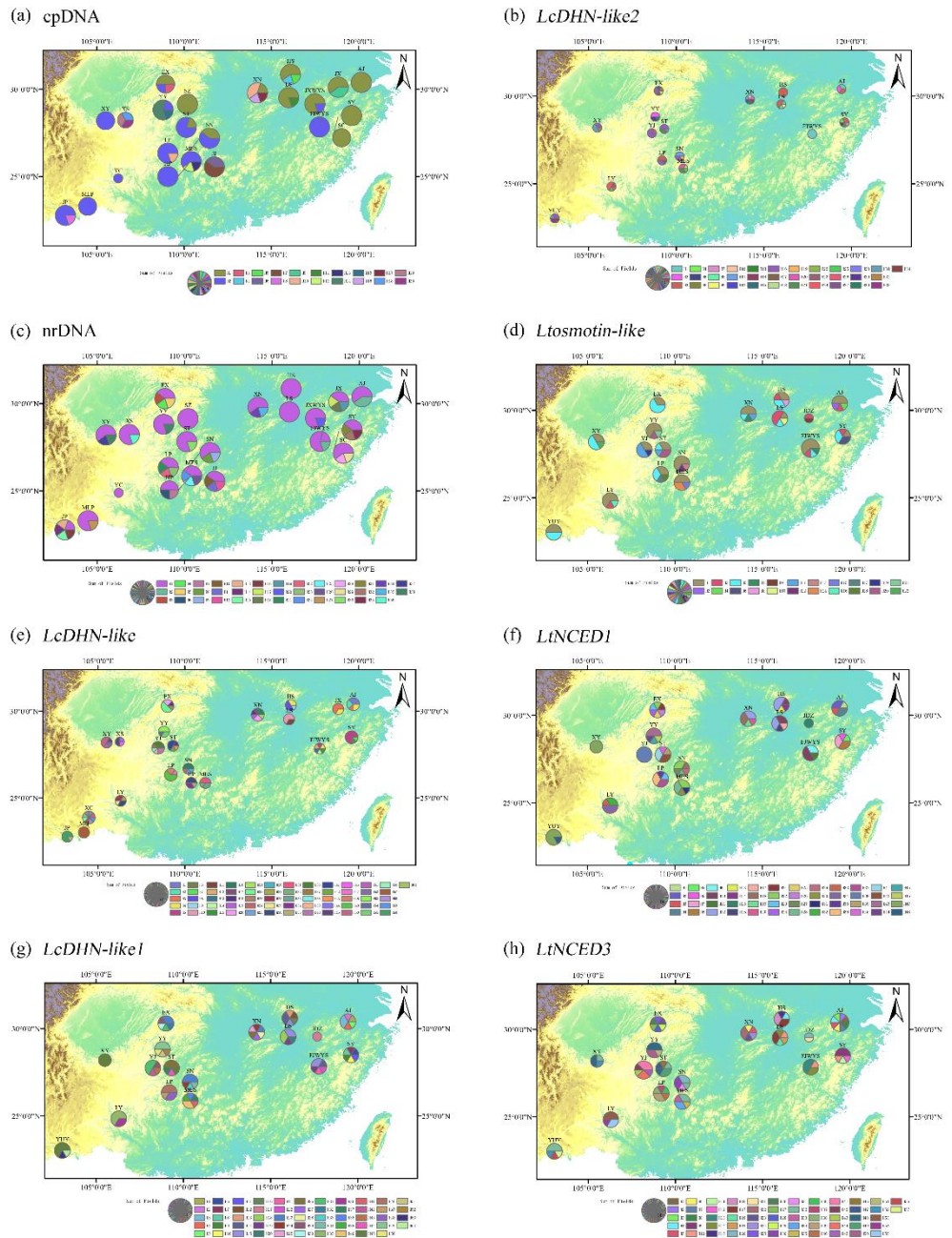

**Figure 5.** Geographic distribution of haplotypes of cpDNA, nrDNA, and six nuclear genes (*LcDHN-like*, *LcDHN-like1*, *LcDHN-like2*, *LtNCED1*, *LtNCED3*, and *Ltosmotin-like*) in *L. chinense* (**a**–**h**). Pie chart size is proportional to its numbers. The map was created by ArcGIS 10.2.

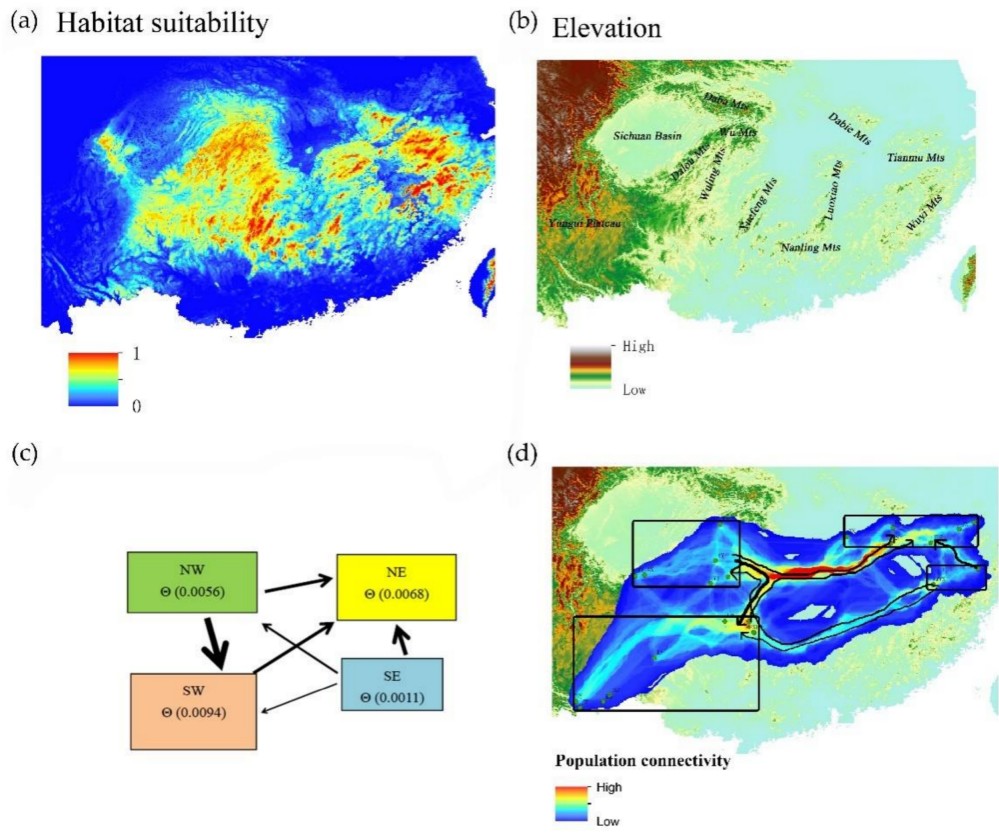

**Figure 6.** Gene flow estimates for *L. chinense*. Arrow width depicts the extent of gene flow between populations. (**a**) "Current" species distribution model of *L. chinense*; warmer colors represent areas of higher habitat suitability. (**b**) Digital elevation model (DEM) and distribution of mainly mountains in subtropical China. (**c**) We estimated gene flow between each pair of the four populations with a coalescent-based framework implemented in MIGRATE 3.6.11, and the numbers in rectangles represent effective population size. (**d**) The results of MIGRATE overlaid onto the dispersal network.

**Table 7.** Marginal likelihood values of all migration models.

| Scenario | Thermodynamic Score | Scenario | Thermodynamic Score | Scenario | Thermodynamic Score |
|---|---|---|---|---|---|
| NE↔SE | −19,099.38 | NE↔SW | −21,453.97 | SE↔SW | −19,761.13 |
| NE→SE | −19,093.69 | NE→SW | −21,443.74 | SE→SW | **−19,749.71** |
| SE→NE | **−19,092.19** | SW→NE | **−21,442.58** | SW→SE | −19,756.28 |
| NE↔NW | −22,357.96 | SE↔NW | −20,756.09 | NW↔SW | −22,700.17 |
| NE→NW | −22,349.87 | SE→NW | **−20,741.89** | NW→SW | −22,695.12 |
| NW→NE | **−22,349.52** | NW→SE | −20,743.84 | SW→NW | −22,698.26 |

Marginal likelihood values of the most likely scenario are indicated in bold.

**Table 8.** Gene flow between pairs of four populations (NE, SE, NW, and SW).

| Scenario | Migration Rate |
|---|---|
| SE→NE | 401.21 |
| NW→NE | 349.47 |
| SW→NE | 332.92 |
| SE→NW | 107.91 |
| SE→SW | 21.56 |
| NW→SW | 495.87 |

### 3.4. Species Distribution Modeling

The MaxEnt model of *L. chinense* showed perfect simulation results. The AUC values for prediction in the five distinct periods were above 0.95 and higher than the value (0.5) for the random model. Generally, an AUC > 0.9 indicates that the simulation result is accurate [41]. The results of the jackknife test showed that Bio17 (precipitation of the driest quarter) made the highest percent contribution (65.3%) of the 12 environmental factors (Table 3). The cumulative percent contribution of the climate variables to the spatial distribution of *L. chinense* was 90.9%, while the contribution of topographic variables was only 9.1% (Table 3).

Currently, *L. chinense* exhibits an island-like distribution in southeastern China; and a banded distribution in southwestern China [27]. As expected, the current distributional predictions (Figure 7a) accurately represented the extant distribution of *L. chinense*, except that no occurrence was predicted in Taiwan. This result suggested that Taiwan may be suitable for this species, and the species may also have been distributed in Taiwan throughout geological history. Relevant evidence needs to be further sought. Paleodistribution modeling suggested that *L. chinense* did not migrate southward in the LGM (Figure 7c). Compared with the LIG (Figure 7b), the LGM exhibited a slightly contracted distribution range of suitable areas, a slightly expanded distribution range of low-suitability areas, and the same distribution range of unsuitable areas. In other words, the original suitable areas were converted to low-suitability areas during the LGM. Compared with the LGM, the MH exhibited a slightly expanded distribution range of suitable areas (Figure 7d), a contracted distribution range of low-suitability areas, and a distinctly increased distribution range of unsuitable areas.

Compared with the distribution ranges of the three historical periods (LIG, LGM, and MH), the current distribution range of *L. chinense* is significantly contracted. In general, the spatial distribution of *L. chinense* has been gradually shrinking since the LIG, and the distribution patterns of the three historical periods were consistent with the current distribution pattern (Table 9).

**Table 9.** The area of suitable area of *L. chinense* in five periods.

| Period | Category | Area ($10^5$ km$^2$) |
|---|---|---|
| Current | unsuitable | 89.51 |
| | low suitability | 3.99 |
| | suitable | 2.68 |
| LIG | unsuitable | 87.65 |
| | low suitability | 4.55 |
| | suitable | 3.86 |
| LGM | unsuitable | 87.67 |
| | low suitability | 5.01 |
| | suitable | 3.50 |
| MH | unsuitable | 88.56 |
| | low suitability | 4.09 |
| | suitable | 3.52 |
| Future | unsuitable | 88.10 |
| | low suitability | 4.51 |
| | suitable | 3.56 |

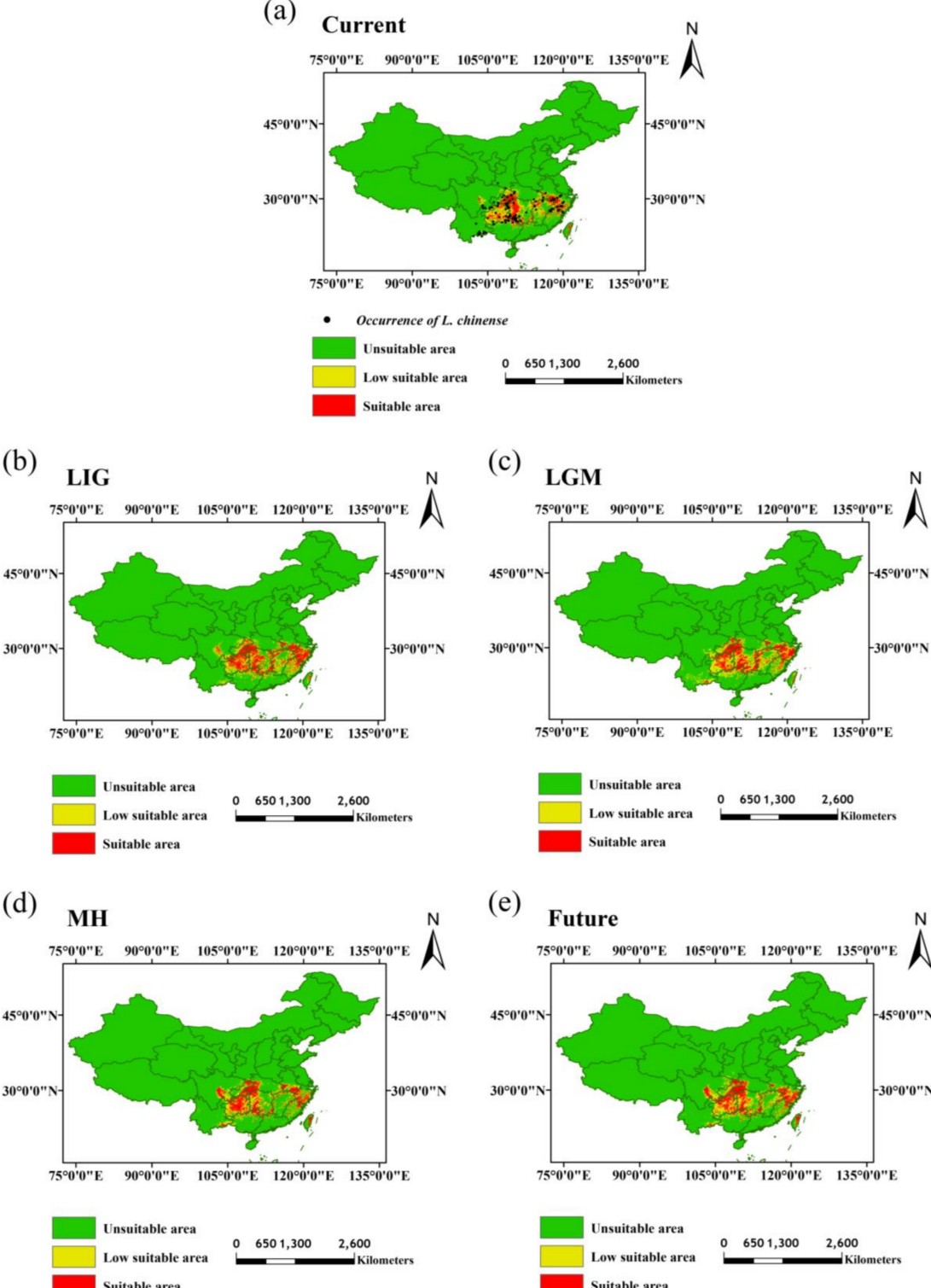

**Figure 7.** Comparisons of current, past, and future distributions of *L. chinense* projected by species distribution model (SDM) using MaxEnt. (**a**) Current distribution (black dots represent data points used in the modeling). (**b**) LIG (Last interglacial) distribution. (**c**) LGM (Last glacial maximum) distribution. (**d**) MH (Mid-Holocene) distribution. (**e**) Future (2080) distribution.

## 4. Discussion

### 4.1. The Among-Population Dispersal Corridor Function of Mountains

There are multiple mountains and hills with different layouts in subtropical China that provide topographic heterogeneity and complex habitats for plants (e.g., *Cathaya argyrophylla*, *Cunninghamia konishii*, *Dysosma versipellis*, *Eurycorymbus cavaleriei*, and *Ginkgo biloba*) [6,7,74]. These mountains play an important role in facilitating gene flow among plant populations [16,48,75]. In this study, we integrated a haplotype network and SDMs and found two obvious dispersal corridors between eastern and western populations. Additionally, we detected dispersal corridors between northern and southern populations. The Nanling Mountains served as a dispersal corridor connecting the eastern and western populations, as shown for other species (e.g., *Eomecon chionantha*, *Eurycorymbus cavaleriei*, and *Pinus kwangtungensis*) [16,75]. In addition, we found a new dispersal corridor connecting the eastern and western populations by intermediates (the northern Xuefeng Mountains and Luoxiao Mountains), which served as stepping-stones between the Wu Mountains and Tianmu Mountains (Figure 6d). The north–south-trending Xuefeng Mountains and Wuyi Mountains served as stepping-stones between the southern and northern populations (Figure 6d). As revealed by previous studies [16,48,75], the Nanling Mountains, Daba Mountains, and Qinling Mountains are distributed in an approximately east–west layout, facilitating the east–west migration of subspecies or related species (e.g., *Ostryopsis davidiana*, *Juglans manshurica*, *Eomecon chionantha*, and *Eurycorymbus cavaleriei*), while the Wuyi Mountains, Luoxiao Mountains, and Wuling Mountains are south–north trending, facilitating the south–north migration of subspecies or related species (e.g., *Platycrater arguta*) in the subtropical region of China (Figure 6b). In addition, nonzero historical gene flow was inferred in six population pairs, and asymmetric bidirectional gene flow was detected between the eastern and western populations (Figure 6c–d). These results implied that the mountains of subtropical China acted as a bridge in the exchange of genetic resources among populations in different regions. Therefore, this study further validated previous conclusions that east–west-trending and north–south-trending mountains played a key role in facilitating migration of many species in subtropical China [16,48,75]. Furthermore, this also implies that the complex and diverse landscape features of subtropical China play an important role in the spatial distribution of genetic variation in plants in this region.

The distribution of haplotypes of cpDNA, nrDNA, *LcDHN-like2*, and *Ltosmotin-like* revealed haplotypes, shared by many eastern and western populations (Figure 5a–d). In addition, the NJ tree based on cpDNA, nrDNA, ITS, and nDNA sequences clustered most eastern populations into a clade separated from the western populations, but minority populations arose from admixture between the eastern and western populations. These results also suggested historical gene flow between the eastern and western populations of *L. chinense*. Furthermore, the dispersal corridors and historical gene flow detected here also provided valuable information about the causes of introgression between the eastern and western populations.

We also estimated the average substitution rates of six nuclear genes. The average substitution rates of the six genes were $2.0 \times 10^{-8}$ s/s/y (*LcDHN-like*), $1.3 \times 10^{-8}$–$1.9 \times 10^{-8}$ s/s/y (*LcDHN-like1*), $8.5 \times 10^{-9}$–$1.3 \times 10^{-8}$ s/s/y (*LcDHN-like2*), $1.7 \times 10^{-8}$–$2.5 \times 10^{-8}$ s/s/y (*LtNCED1*), $2.0 \times 10^{-8}$–$2.9 \times 10^{-8}$ s/s/y (*LtNCED3*), and $8.0 \times 10^{-9}$–$1.2 \times 10^{-8}$ s/s/y (*Ltosmotin-like*). The average substitution rates of four genes (*LcDHN-like*, *LcDHN-like1*, *LtNCED1*, and *LtNCED3*) were higher than those of the remaining two genes (*LcDHN-like2* and *Ltosmotin-like*). These genes with high substitution rates may have accelerated the divergence between the eastern and western populations and caused the disappearance of dispersal corridors between them (Figure 4e–h). Previous studies using mitochondrial and nuclear genes of animals showed similar results [39,49].

### 4.2. Isolation of Glacial Refugia and Differentiation between Eastern and Western Populations

Identifying the past, present, and future distributions of species can not only help understand the potential distribution areas of species, reconstruct the historical distributions of species, and identify

glacial refuges but also enable the formulation of reasonable resource conservation and management strategies for endangered species. The distribution of *L. chinense* in China is largely dependent on bioclimatic variables, and the influence of topographical variables is relatively small (Table 3). This result was consistent with that of previous studies showing that climate factors play a more important role than topographical factors in determining species distributions at a large geographical scale [76,77]. Among 10 bioclimatic factors, Bio17 (precipitation of the driest quarter) exhibited the greater percent contribution to the distribution of *L. chinense*, suggesting that the Bio17 climatic factor played a key role in determining the distribution of *L. chinense*. This result is consistent with that of previous studies showing that *L. chinense* is very sensitive to water conditions [78,79].

The SDM results showed that the current distributional predictions accurately represented the extant distribution of *L. chinense*. The eastern populations were distributed in the Dabie Mountains, northern of Luoxiao Mountains, Tianmu Mountains, and Wuyi Mountains, while the western populations were distributed along the Xuefeng Mountains, Daba Mountains, Wu Mountains, Dalou Mountains, Wuling Mountains, and southeastern Yungui Plateau (Figure 6a). The distributions during the paleoclimatic periods (LIG, LGM, and MH) were consistent with the current distribution (Figure 7a–d), suggesting the presence of multiple refuges in multiple isolated mountain regions in China, an in situ refugia pattern. In previous studies, similar to *L. chinense*, many species (e.g., *Juglans mandshurica*, *Davidia involucrate*, and *Euptelea pleiosperma*) in China's subtropical regions showed an in situ refugia pattern [8,9,80]. These mountains also served as glacial refuges for many species (e.g., *Quercus glauca*, *Cathaya argyrophylla*, *Eurycorymbus cavaleriei*, *Ginkgo biloba*, and *Pinus kwangtungensis*) in subtropical China [6,81,82]. Species in mountain regions are less affected by climatic fluctuations than those on the plains, and they have been able to migrate to different suitable elevations in mountains during Quaternary glacial and interglacial periods. Therefore, these mountains generally support abundant species and serve as refugia for many Tertiary relic species (e.g., *Emmenopterys henryi*, *Tetracentron sinense*, *Cyclocarya paliurus*, and *Cercidiphyllum japonicum*) due to their heterogeneous and complex environments [5–7,74]. These studies further suggested that the mountains in subtropical China play a key role in the conservation of genetic resources of species. Therefore, these mountains should be served as conservation centers for species genetic resources, which contribute to the conservation of species genetic diversity in subtropical China.

The long-term in situ refugia pattern of *L. chinense* populations may have caused genetic differentiation among isolated populations. In this study, the NJ tree based on cpDNA, nrDNA, and nDNA sequences clustered most eastern populations into a clade separated from the western populations, which was consistent with the pattern of their natural geographical distribution [25–27]. The barrier analysis also revealed genetic barriers between the eastern and western populations (Figure 3a–b). In addition, previous studies reported distinct east–west lineage divergence in *L. chinense* [31,33,34]. Yang used cpDNA sequence variation information and inferred that the divergence time of the east–west lineage was 0.443 Ma ago [31], earlier than the LIG (~0.12–0.14 Ma). Thus, the expansion and migration of *L. chinense* populations might have ended before the LIG. In addition, the smaller amount of historical gene flow between eastern and western populations than within eastern western populations suggests the intensification of genetic differentiation between the eastern and western populations (Table 7).

Previous studies of fossil pollen showed that the distribution ranges of temperate deciduous forests and mixed temperate-boreal forests were widespread in subtropical China (22° N–30° N) during the LGM [6,74]; currently, these regions are inhabited by warm-temperate evergreen forests, and temperate deciduous forests have migrated to higher-latitude regions in China [6,74]. In other words, temperate deciduous forests may retreat to lower-latitude regions during cooler glacial periods and return to higher-latitude regions during warmer interglacial periods. Frequent phytocoenosis succession events occurred with climate fluctuations in the Quaternary, which resulted in habitat fragmentation of some temperate deciduous forest species [6].

As a temperate deciduous tree, *L. chinense* may have been affected by Quaternary climatic fluctuations. Repeated glacial and interglacial cycles caused *L. chinense* to retreat to its currently fragmented refuges [32]. These refuges are located in different mountain regions, which provide diversified, suitable habitats for *L. chinense* [32]. Therefore, we assume that eastern and western populations were able to migrate and disperse through the two dispersal corridors before the LIG or even farther back in time. Subsequently, due to climatic fluctuations of the repeated glacial and interglacial periods in the Quaternary, migration corridors and most of the habitats were frequently occupied by warm-temperate evergreen and mixed temperate-boreal forests [6,74]. Finally, the eastern and western populations gradually diverged under the condition of long-term isolation, forming the current geographical distribution pattern. In this study, the distribution of haplotypes of cpDNA, nrDNA, *LcDHN-like2*, and *Ltosmotin-like* revealed haplotypes shared by many eastern and western populations (Figure 5a–d). In contrast, the haplotype networks of *LcDHN-like*, *LcDHN-like1*, and *LtNCED3* separated the eastern populations from the western populations (Figure S1). In addition, the NJ tree based on cpDNA, nrDNA, and nDNA sequences clustered most eastern populations into a clade separated from the western populations, but a few populations arose from admixture between the eastern and western populations. There was no significant difference in genetic diversity between the eastern and western populations (Tables 4 and 5). These results suggested that genetic divergence between eastern and western populations was most likely a gradual process. Furthermore, due to endangered habitats, intense interspecific competition, low seed viability, and artificial interference, *L. chinense* populations have been restricted to their current refuges [25,26]. These may further exacerbate the genetic differentiation between the eastern and western populations. The SDMs predicted a slightly expanded distribution of *L. chinense* in 2070 (2060–2080) compared to the current range (Figure 7a). Thus, global warming might not greatly threaten the survival of *L. chinense* in terms of natural habitat suitability in the next 50 years. In addition, with the advances and development of high-throughput sequencing technology and the decline in sequencing costs, it is more feasible to obtain individual or population genomic data [83–85]. DArT-seq, RAD-seq, and genome-wide association study (GWAS) have shown to be cost-effective methods for generating genome-wide DNA markers for a large number of samples for phylogeographic and population genomic studies [43,84,86]. Therefore, combining high-throughput technology with landscape ecology will be more powerful for providing reasonable resource conservation and management strategies for more species.

## 5. Conclusions

In this study, two dispersal corridors were detected between eastern and western populations of *L. chinense*, and these dispersal corridors were located in mountain regions. Potential historical gene flow and admixture events of minority populations between the eastern and western populations indicated the occurrence of migration between the eastern and western populations during their evolutionary history. The SDM results suggested that the distribution range of *L. chinense* has been shrinking since the LIG, and showed an in situ refugia pattern in multiple mountain regions. The genetic divergence between the eastern and western populations was revealed by NJ trees of cpDNA and nDNA.

Due to climatic fluctuations in multiple glacial and interglacial periods, in the Quaternary, dispersal corridors and habitats were frequently inhabited by warm-temperate evergreen forests and mixed temperate-boreal forests [6,74], which may have exacerbated the differentiation of the eastern and western populations and result in the fragmentation of *L. chinense* habitats. These findings suggested that the genetic divergence between eastern and western populations was most likely a gradual process. The topographic heterogeneity and complex environments of mountainous regions [5–7] provide suitable habitats for *L. chinense*. Therefore, these dispersal corridors and mountainous refugia suggested that the mountains in subtropical China play a crucial role in the conservation of genetic resources and migration of subspecies or related species in this region. Furthermore, the study also provides a reference for the study of other species endemic to subtropical China.

**Supplementary Materials:** The following are available online at http://www.mdpi.com/1999-4907/10/7/565/s1. Table S1: List of 145 present records of *L. chinense*. Table S2: List of 21 environment variables used in this study. Figure S1: Haplotype networks of cpDNA, nrDNA, and six nuclear genes (*LcDHN-like*, *LcDHN-like1*, *LcDHN-like2*, *LtNCED1*, *LtNCED3* and *Ltosmotin-like*) in *L. chinense* (a–h).

**Author Contributions:** H.L. designed the outline of this paper. Y.C. and K.L. collected the molecular data. Y.S. completed the analysis and calculation using the different methods and software. Y.S. wrote the paper. Y.C. and H.L. gave some advice in the manuscript.

**Funding:** This study was supported by the National Natural Science Foundation of China (No. 31770718 and No. 31470660).

**Acknowledgments:** We are thankful for funding from National Natural Science Foundation of China (No. 31770718 and No. 31470660) and the Priority Academic Program Development of Jiangsu Higher Education Institutions (PAPD).

**Conflicts of Interest:** The authors declare no conflict of interest.

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
