# Peer review of "Integrating Phylogeographic Analysis and Geospatial Methods to Infer Historical Dispersal Routes and Glacial Refugia of Liriodendron chinense"

_forests, doi:10.3390/f10070565_

Round 1
Reviewer 1 Report
This is an interesting manuscript that is strong technically and very well written. I have a few concerns that can be addressed in revisions and do not affect the paper’s foundation of study design, data, and analyses. Most importantly, the manuscript is hyper-focused on the study species to the point of neglecting the context of conservation, understanding and management of regional forests, and landscape genetics more generally; this is especially true in the Discussion section. More specific concerns are addressed below, by figure or manuscript line number.
Abstract. I think that the abstract includes more detail on methods than is necessary. Summarizing methods more would allow a little more room for focusing on implications. See note for lines 26-28.
20, 25. The authors use the term “obvious” twice in the abstract. In my interpretation, the usage trivializes the results and should be eliminated.
26-28. This reflects a narrow focus on the intrinsic interest of the study species and misses an opportunity to mention broader implications. Does the loss of these dispersal corridors due to glacial cycles explain the present distribution of diversity? Does this have implications for the conservation of the study species? Does this explain more about the composition of forests in subtropical China or have implications for landscape genetics principles?
126-129. This is true, but a little more detail should be mentioned, since the sample sizes seem very small compared to the long history of marker-based genetic studies.
165. The quote here is jarring, as it does not transition well to the following sentences and is out-of-place regarding other mentions of GenGis. If this is meant to be an introductory sentence, I suggest removing the quote and developing a sentence that really summarizes the activities.
191-198. A concern with species distribution models, especially for rare or endangered species, is that the present distribution is often much smaller than the potential distribution, as it has been affected by habitat destruction or related phenomena, or by harvesting, leading to bias in the model. So, it would help to have more information about the occurrence records used—do they reflect only current distribution (as suggested by the wording), or do they include historical or fossil records? The supplemental table listing these records was not available for this review, so I do not know if they would resolve my questions here.
348. What are “perfect simulation results”? Does this mean that the results were as expected, or is there a technical definition that is relevant to the MaxEnt analysis? In the former case, this wording should be removed, as it implies an investment in results; in the latter case, the technical definition should be clarified.
354-356. The figure reference should be to figure 7a exclusively, since that is the figure that shows occurrence together with predicted distribution. Since the current prediction is modeled from the occurrence records, it seems redundant to state that they are in agreement. If this is a validation, perhaps that should be stated (e.g., “as expected”). The statement that “no occurrence is predicted in Taiwan” appears to conflict with figure 7a/e, in which there is a small but obvious red area on the map of Taiwan.
Figure 5. As far as I can tell, this figure is mentioned only to illustrate that some haplotypes are shared across the east/west divide (292-311). The figure as presented is probably not the most effective way to demonstrate that: could a statistical summary (bar graph, or pie chart over a map, perhaps) be used to indicate the presence of east/west haplotypes, either summarized for all loci or by individual locus? The other potentially important item in the haplotype networks is the fact that the haplotypes are not well segregated and therefore are difficult to explain by single colonization events or prolonged isolation with in situ evolution of haplotypes… but since this is not a major point of the paper the networks themselves could probably be a supplemental figure.
Figure 7. The green triangles are very difficult to see against the green “unsuitable area” regions in 7a on some displays. If possible (depending on software used), different symbols or a different color scheme should be used for 7a. Since the occurrence records are shown only in (a), this should be reflected in the legend, which would also explain why essentially the same image is shown twice (7a/e). Also, in figure 7a/e, the identified areas of suitability for L. chinense is very small and reinforces my concerns about limitations of the sampling/occurrence records used for the SDM’s, as described in my comments for lines 191-198. Is it realistic to expect that the recent distribution of the species is so limited, and not an artifact of habitat destruction, harvesting, or other factors?
Discussion, general: The exaggerated focus on the target species, L. chinense, is especially obvious in the Discussion. For example, I do not see any other species mentioned by name, and only vague references to other forest species in subtropical China. As an endangered species, I understand that there is inherent value in understanding the microevolution of this species, but I would like to see more discussion of the context and implications.
403. Incomplete lineage sorting is not the same as introgression/admixture by gene flow; they are distinct phenomena that can produce similar results. I would like to see this clarified with references.
Reviewer 2 Report
This is a well presented manuscript and is of significant interest to readers, not only in reference to Liriodendron chinense but also to the use of a multidiciplinary approach to conservation management.
In my opinion, the article is worthy of publication, however a number of details need clarification and/or additional work.
Major Comments
Line 16: This is a small number of loci and small amount of data relative to current molecular methods. In the methods and/or discussion sections please justify why/how six loci provides enough resolution power to interpret genetic structure and please highlight other techniques that could be used such as DArT, SSRs, RAD-SEQ etc. See also, Major Comments 4 & 5.
Section 3.1: I found this to be an unusual way to analyse genetic diversity. This analysis seems to be locus to locus, whereas most studies would assess genetic diversity population by population. Can you explain why you have conducted the analysis this way? And pending your assessment after considering my request, potentially provide a population by population genetic diversity analysis.
Line 288 & 484: The analyses relating to genetic divergence do not appear to provide any scale or statistical analysis. Can you provide data/analyses to justify the claim of 'significant' genetic divergence? Also, how does this relate to the lack of significant difference in nucleotide diversity reported in Section 3.1?
Section 4.2: What other types of molecular markers could you have used for this study, and why are the markers you have chosen appropriate? Are there specific benefits of the type of markers you have chosen compared to other options? Could the use of other markers have provided any additional important data? In my opinion, you need to add commentary addressing this. This question clearly aligns very closely with Major Comment 5 and Major Comment 1.
Section 4.2: Can you please discuss how the genetic data you present in this paper provides additional information not already in the literature? Yang et al, 2019 as well as Zhong et al 2019, seem particularly relevant; Zhong et al uses RAD-Seq, which may be a more powerful tool for genetic analysis as compared to the technique utilised by your team. Can you please clearly discuss why the genetic data you provide is an advance on current literature and incorporate in more detail previous literature and outcomes. This question clearly aligns very closely with Major Comment 4 and Major Comment 1.
Line 477: Your claim of 'we found', needs to be justified within the manuscript. There is much literature on this topic. Please refer to Major Comment 5.
Minor Comments
Line 38: Insert 'which' or 'that' before 'often'.
Line 87-99: The use of a multidiciplinary approach to provide conservation management advice is very powerful and should be 'best practice' for all such studies. But yours is not the first article to use this approach and indicate its importance. Therefore, within this section (or elsewhere as you see fit), Conroy et al 2019 (https://doi.org/10.1371/journal.pone.0210560) would seem appropriate to reference.
Line 125: What was the size of each nominal population? Were they similar in size/geography etc? How far apart were the samples in each population? Were the distances between samples similar population to population? Were the samples all adults, the same size etc?
Line 127 & 128/129: Do you mean 'samples per population'?
Line 450: This is a bit ambiguous. What does the word 'less' connect with, historical or gene flow?
Round 2
Reviewer 2 Report
Thank you for your detailed responses.
Regarding point 1. I think your response is fair enough and whilst I am not 100% sure I agree, you have provided appropriate justification and discussion within the manuscript, and I therefore think this is appropriate. However, you refer to 'population genetics', most significantly within the title and abstract but elsewhere too. In my opinion, the analysis you have undertaken using cpDNA and nrDNA markers do not facilitate a population genetic study; really those markers are better suited for phylogenetic analysis, which to me implies multiple species. Your own information highlights the use of these markers in phylogeographic analysis. Therefore, I request that you modify reference to population genetics to phylogeographic analysis as in my opinion, reference to population genetics is misleading to potential readers.
Re point 5. Yes, OK that's reasonable. As per point 1 above, I am not 100% sure I completely agree, but as long as you discuss and justify, readers can use their own judgement. On that note, from what I can tell, you haven't modified the text in relation to this. There isn't much point just justifying it to me as part of the review, you need to add some brief text identifying the other analyses and why your data adds to this. Have you referenced Zhang et al 2019? You should.
Re point 9. Thats all fine. But there is no point telling me, you need to add detail to the manuscript for readers to read.